# Joint Encryption Model Based on a Randomized Autoencoder Neural Network and Coupled Chaos Mapping

**DOI:** 10.3390/e25081153

**Published:** 2023-08-01

**Authors:** Anqi Hu, Xiaoxue Gong, Lei Guo

**Affiliations:** 1School of Communication and Information Engineering, Chongqing University of Posts and Telecommunications, No. 2, Chongwen Road, Nanan District, Chongqing 400065, China; anqi3699@gmail.com (A.H.); 2023guolei@gmail.com (L.G.); 2Institute of Intelligent Communication and Network Security, Chongqing University of Posts and Telecommunications, No. 2, Chongwen Road, Nanan District, Chongqing 400065, China

**Keywords:** AENN randomization, coupled chaos mapping, one-time pad, joint encryption, data encryption

## Abstract

Following an in-depth analysis of one-dimensional chaos, a randomized selective autoencoder neural network (AENN), and coupled chaotic mapping are proposed to address the short period and low complexity of one-dimensional chaos. An improved method is proposed for synchronizing keys during the transmission of one-time pad encryption, which can greatly reduce the usage of channel resources. Then, a joint encryption model based on randomized AENN and a new chaotic coupling mapping is proposed. The performance analysis concludes that the encryption model possesses a huge key space and high sensitivity, and achieves the effect of one-time pad encryption. Experimental results show that this model is a high-security joint encryption model that saves secure channel resources and has the ability to resist common attacks, such as exhaustive attacks, selective plaintext attacks, and statistical attacks.

## 1. Introduction

With the gradual development of cryptanalysis and the continuous improvement of computer processing speeds, certain weaknesses of traditional cryptography are gradually being exposed [1]. In order to ensure the secure storage and transmission of information in cloud computing, big data, and other new fields, there is an urgent need to research and design new cryptographic technology and theory. In recent years, chaotic cryptography, as a new encryption technology, has attracted the attention of researchers in various fields at home and abroad [2,3,4]. The two major research directions of chaotic cryptography are stream ciphers and block cipher systems based on chaos theory, and chaos synchronization-centered secure communication systems. Many encryption algorithms based on chaos have been proposed and applied [5,6,7,8,9]. The certainty and unpredictability of chaotic systems meet the basic requirement of information encryption, which is information hiding and recovery [10]. In addition, the sensitivity of chaos to initial conditions conforms to the diffusion characteristics of cryptography [11], and the pseudo-randomness conforms to the confusion characteristics of cryptography [12]. These two characteristics allow chaotic stream encryption to naturally have a superior information-hiding function.

However, early studies also proved that chaotic stream encryption has obvious drawbacks, such as the poor resistance of low-dimensional chaotic encryption systems to exhaustive attacks [13,14,15]; the equivalent keys in designing the encryption system, which cannot effectively resist known plaintext attacks [16,17], etc. Therefore, the traditional chaotic stream cryptography needs to be improved to enhance security. Continuous-time high-dimensional hyperchaotic systems have been proposed [18,19] encryption security has greatly improved because it has higher dimensionality and more complex dynamics, and the iterative equations have more initial values and control parameters. To continuously improve encryption security, researchers are constantly striving to find new high-security methods to participate in the encryption step, such as the encryption method based on a neural network. Among them, the typical one is the neural network [20,21] constructed by a memristor, which not only has behavior similar to a chaotic system, but can also form a complex neural network model, produce rich discharge patterns, and have high security. Ref. [22] designs a flexible image encryption algorithm by using BP neural network compression technology and chaotic mapping and proves its security. From many practical and forward-looking encryption algorithms, it can be seen that an encryption algorithm combining multiple steps and algorithms offers higher security than a single algorithm.

With further demands for security performance, combinations of the one-time pad and chaotic encryption are also being proposed. It has been proven that the one-time pad technique has strong security [23,24,25,26]. The one-time pad chaotic stream encryption proposed by [23,27] is similar in nature to the one-time pad proposed by Shannon, which requires real-time sharing of keys through secure channels or management of complex key systems. However, this way of sharing keys in real time adds additional communication burdens, especially for short messages.

In view of the disadvantages of the chaotic encryption schemes mentioned above, such as insufficient security or low encryption efficiency, two low-dimensional chaotic coupling schemes are proposed to relatively balance security and encryption efficiency. Since the autoencoder neural network (AENN) has enough nonlinear complexity to make up for the shortage of nonlinear complexity in chaotic stream encryption, a joint encryption model based on a randomized AENN and improved chaotic coupling map is proposed.

The contributions of this paper are summarized as follows:Based on the auto-learning of AENN, an AENN encryption model through randomized selection is constructed, which enlarges the non-linear complexity and key space of the encryption algorithm.Aimed at the key shortage in a one-dimensional chaotic space and the complexity of high-dimensional chaotic operations, a new chaotic map coupled with improved Chebyshev map (ICM) and improved logistic map (ILM) is designed.In view of the large amount of channel resources consuming the traditional one-time pad encryption method, the iterative data of a chaotic map are used to perform another iteration and combine the number of communications to comprehensively calculate a new initial condition of the chaotic map, so that the system can share a key for the first time and the key can be continuously changed and transmitted. After the verification of the encryption experiment, we can conclude that the joint encryption model that we propose is a high-security encryption model that saves the one-time pad channel resources and can resist common attacks.

The principles of AENN, ICM, and ILM are introduced in Section 2; in Section 3, the implementation of AENN randomization and the new coupled chaos mapping is described; in Section 4, the encryption and decryption algorithm of the joint encryption model is described; in Section 5, the security performance of the joint encryption model is analyzed from the perspectives of key space, key sensitivity, and resistance to selective plaintext and statistical attacks; in Section 6, the joint encryption model is used to encrypt text and images, respectively, and the experimental results are analyzed; Section 7, summarizes the innovation and its effects and proposes further improvements.

## 2. Related Work

### 2.1. Improved Chaos Map

A nonlinear system, which is highly sensitive to small changes in initial conditions, is called a chaotic system. A chaotic system has a high sensitivity to control parameters, good pseudo-randomness, ergodicity, and long-term unpredictability, which are qualities that are similar to the confusion and diffusion characteristics found in cryptography [28]. Using chaotic dynamic behavior to diffuse and confuse plaintext has an avalanche effect, so chaotic stream encryption has more advantages than traditional encryption methods. The security of chaos is related to the complexity of chaotic systems. Common chaotic maps are logistic, sine, tent, and so on [29,30]. These one-dimensional chaotic maps offer small key spaces and limited resistance to violent exhaustive attacks, and some chaotic maps have obvious short-period phenomena, making them vulnerable to statistical attacks [31]. To address the above drawbacks of chaotic encryption, an ICM and an ILM are adopted for cross-mapping to improve the key space and spacetime complexity of chaotic systems, greatly improving the security of chaotic encryption.

In order to expand the range of the control parameter μ and increase mapping complexity, in [32], the μ value is modified and an ICM is defined, as follows:(1)$$\begin{array}{c} U_{k+1}=\left\lceil \cos\left(\left(\mu +1\right)\cos^{-1}\left(U_{k}\right)\right)\times 2^{15}\right\rceil \\ -\cos\left(\left(\mu +1\right)\cos^{-1}\left(U_{k}\right)\right)\times 2^{15} \end{array}$$
where, the parameter μ∈(0,10], $U_{k}\in \left[0,1\right]$.

The chaotic bifurcation diagram and Lyapunov exponent (LE) diagram of the Chebyshev map and ICM [32] are shown in Figure 1. The bifurcation diagram shows the abnormal behavior of the Chebyshev mapping μ value between 1 and 2, while the ICM exhibits chaotic behavior throughout the entire parameter range. It can also be seen from the LE diagram that the LE of the original Chebyshev map is less than 1 when μ<1, indicating that the original Chebyshev map is not in a chaotic state when μ<1. The LE of ICM is greater than 1 at μ∈(0,10), which indicates that ICM is in a chaotic state at μ∈(0,10). The maximum LE of ICM at μ∈(0,10) is 12.63, which is larger than the maximum LE of the original Chebyshev map (2.30), making ICM more secure than the Chebyshev map.

One-dimensional logistic maps can easily generate a large number of sequences quickly, but their low complexity results in weak security performance. So an ILM is designed in [33], which is defined as follows:(2)$$X_{k+1}=1.0-aX_{k}{}^{2}-\left|X_{k}\right|$$
where the initial value $X_{0}\in \left(-1,1\right)$.

The premise of cross-coupling mapping of different chaos necessitates that they share the same range of values. The range of ICM is $U_{k}\in \left[0,1\right]$ (special treatment applies for $U_{k}=0$ or $U_{k}=1$), so we calculated the absolute value of ILM, so that $X_{0}\in \left(0,1\right)$; Equation (2) can be expressed as follows:(3)$$X_{k+1}=\left|1.0-aX_{k}^{2}+\left|X_{k}\right|\right|$$

Figure 2 shows the bifurcation diagram and LE diagram of the logistic map and ILM. According to the comprehensive analysis of the bifurcation diagram and LE diagram, the logistic map is in a chaotic state when a∈(3.6,4), and ILM is in a chaotic state when a∈(0,1). When ILM is at a∈(0,1), the maximum LE is 0.6812, which is greater than the maximum LE of the logistic map (0.6586). Moreover, the LE of ILM in the whole value range of a is greater than 0, and it has a wider range of chaotic behavior, and its overall performance is better than that of the logistic map.

The sequence generated by chaotic mapping iteration is a real number sequence and cannot be directly used for encryption. It is necessary to quantize the real number sequence to obtain the bit sequence. The commonly used quantization methods include binary quantization, multiple coarse-graining, integer remainder quantization, multi-level uniform quantization, and so on. The chaotic sequence quantized by multi-level uniform quantization has good uniformity and correlation characteristics, and it is difficult to reconstruct it by inverse iteration. Moreover, each iteration of the chaotic function can obtain multiple bits, which can speed up the generation of the sequence. To obtain a uniform bit sequence, the L power of two is usually used as the quantization coefficient, so the expression of transforming the chaotic real number sequence $\left\{x_{k}\right\}$ into the integer sequence $\left\{y_{k}\right\}$ is as follows:(4)$$y_{k}=\left\lfloor 2^{L}x_{k}\right\rfloor$$
where $\left\lfloor \cdot \right\rfloor$ is used for rounding down. The obtained integer sequence is converted into equally long binary sequences, which are then concatenated together to form a binary bit sequence that can be used for encryption.

### 2.2. Autoencoder Neural Network (AENN)

Neural networks are used to identify and process potential representations in digital signals by simulating the operation of the human brain [34,35]. AENN is an unsupervised neural network model, which usually consists of an encoder and a decoder. Unlike other encoders (ENNs), the task of the autoencoder is to map itself. In theory, the AENN encoder attempts to map the original data with smaller dimensions without losing data information. The AENN decoder reconstructs the encoded low-dimensional data to restore the original data. In this paper, the AENN encoder is used to encode the data, and the decoder is used to reconstruct the data, thus realizing the encoding and decoding of the data. In order to improve the encryption efficiency under the condition of complete decoding, this paper follows the principle of simplicity and effectiveness and chooses an AENN model with the least parameters and the simplest structure for encryption. The AENN architecture is shown in Figure 3.

As shown in Figure 3, the network structure of AENN includes an input layer, a hidden layer, and an output layer, wherein the input layer and the hidden layer in the middle constitute an encoder (Figure 3a), and the hidden layer in the middle and the output layer constitute a decoder (Figure 3b). It should be emphasized that the hidden layer in the middle serves as both the output of the encoder and the input of the decoder, and the coding function is embodied through the hidden layer. Suppose there are *N* input nodes, the *i*th input node is represented by $x_{i}$, and the *j*th hidden node is represented by $z_{j}$. The weight parameter between the *i*th input node and the *j*th hidden node is marked as $w_{ij}$, and $b_{h(j)}$ represents the bias of the *j*th hidden node. Then $z_{j}$ can be expressed as follows, which is the weighted sum:(5)$$z_{j}=\sum_{i=1}^{N}x_{i}\cdot w_{ij}+b_{h(j)}$$

In the neural network, the value of the hidden node calculated from the upper layer is generally not directly used as the input of the next layer but needs to be transformed by the activation function. The commonly used activation functions are the sigmoid function, ReLU function, and so on. If using the sigmoid function, the hidden node can be represented as follows:(6)$$h_{i}=f\left(z_{i}\right)=\frac{1}{1+e^{-z_{i}}}$$

The ReLU function is a piecewise linear function defined as the positive part of the parameter. If using the ReLU function, the hidden node can be represented as follows:(7)$$h_{i}=\mathrm{ReLU}\left(z_{i}\right)=\left\{\begin{array}{cc} z_{i}, & z_{i}>0\\ 0, & z_{i}\leq 0 \end{array}\right.$$

In order to obtain a reasonable weight value to fit the relationship between the input layer and the output layer, and obtain the output result with a small loss value, it is necessary to establish a reasonable loss function [36] to evaluate the output result. Mean square error (MSE) [37] is a commonly used evaluation function, which can be expressed as follows:(8)$$E=\frac{1}{N}\sum_{i=1}^{N}{\left(y_{i}-x_{i}\right)}^{2}$$

The smaller the MSE, the closer the output of the autoencoder to the input. The training process involves making E→0. Gradient descent (GD) is a commonly used optimization algorithm in deep learning algorithms. When the neural network is initialized, the objective function J(w,b) is not optimal, and the weight values *w* and biases *b* are very small random values close to zero. The gradient of the objective function J(w,b), with regard to parameters *w* and *b*, is the fastest rising direction of the objective function. As long as one steps along the opposite direction of the gradient, the optimization of the objective function can be achieved. By constantly updating weights and biases, we can finally find the point with the least error. The gradient of the objective function J(w,b), with regard to parameters *w* and *b*, can be expressed as follows:(9)$$\begin{array}{c} w_{ij}=w_{ij}-\alpha \cdot \nabla_{w_{ij}}J\left(w,b\right)\\ b_{ij}=b_{ij}-\alpha \cdot \nabla_{w_{ij}}J\left(w,b\right) \end{array}$$

During each update, all gradients need to be calculated on the entire dataset, which will lead to redundancy in the calculation and slow down the speed of training. Thus, to improve the training speed, the gradient can be updated with each training sample, which is the stochastic gradient descent (SGD) method [38]. By using the SGD method, the gradients of the objective function J(w,b) with respect to *w* and *b* can be expressed as follows:(10)$$\begin{array}{c} w_{ij}=w_{ij}-\alpha \cdot \nabla_{w_{ij}}J\left(w,b;x_{i};y_{i}\right)\\ b_{ij}=b_{ij}-\alpha \cdot \nabla_{b_{ij}}J\left(w,b;x_{i};y_{i}\right) \end{array}$$

Among them, α is the learning rate, $\nabla_{w_{ij}}$ is the partial derivative of J(w,b) with respect to $w_{ij}$, and $\nabla_{b_{ij}}$ is the partial derivative of J(w,b) with regard to $b_{ij}$. By constantly updating $w_{ij}$ and $b_{ij}$, the loss function E→0 and the optimized objective function J(w,b) are obtained.

### 2.3. Joint Encryption Model

Due to the openness of the internet and the convenience of data connections, information protection has become an inevitable demand in contemporary social development. Since the information age, people have been exploring information encryption algorithms to protect the storage and secure transmission of data, among which, common typical encryption algorithms are asymmetric, represented by RSA (Rivest–Shamir–Adleman), standard encryption represented by AES (advanced encryption standard), S-box, chaotic encryption, compression encryption, encoding encryption, neural network encryption, etc. These algorithms play an important role in the security and encryption efficiency of different encryption objects and application scenarios to protect data security. The advantages and disadvantages of the above typical encryption algorithms are summarized as shown in Table 1.

From Table 1, it is clear that different single encryption algorithms have different advantages and disadvantages, so the strengths should be used to make up for the weaknesses, using different encryption algorithms to jointly encrypt to achieve the effect of increasing security. For example, a joint encryption scheme is proposed in [39], which encrypts the original image through the improved AES algorithm and the wheel key generated by the chaotic system. This joint encryption scheme not only reduces the time complexity of the algorithm, but also increases the diffusion ability in the algorithm, and has plaintext sensitivity and key sensitivity. The combined encryption algorithm not only enlarges the key space but also has the advantages of the two encryption algorithms, which greatly improve encryption security. In recent years, chaos combined with other encryption algorithms, such as RSA encryption [40,41], neural network encryption [22,42,43], compression encryption [44,45], and encoding encryption [46,47], have further improved security performance.

Generally speaking, a larger key space means that the more the permutation and combination of keys, the greater the difficulty of exhaustive cracking. The combined key spacing of two or more encryption algorithms is larger than that of a single key space. Among them, a special one is neural network encryption. What sets neural network encryption apart from traditional encryption algorithms is that instead of performing mathematical- or algebraic-based operations on the key and data, it uses a pre-set neural network model to continuously train the encryption and decryption processes of the data to obtain an encryption and decryption model with many neuron parameters. Because of the uniqueness of the neuron parameters of each trained model, the neuron parameters can be used as part of the key space, giving the neural network encryption algorithm a massive key space.

Among them, the most typical model is the anti-neural network encryption model proposed by [48]. This model eliminates the possibility for eavesdroppers to use a large number of ciphertexts to train and decrypt the network through countermeasure training. However, the biggest disadvantage of this model is that the decryption model has a certain decryption failure rate. In [49], an improved scheme of this model is proposed. The training model uses two optional keys to encrypt and decrypt data, allowing eavesdroppers to increase the choice and discrimination of keys. The results of countermeasure training give the model resistance to selective ciphertext. However, this model does not solve the decryption failure rate problem. Therefore, using neural network encryption requires solving the decryption failure problem, and using bytes as the unit for AENN training can achieve 100% decryption.

## 3. Realization of AENN Random Selection and New Coupled Chaos Mapping

The AENN randomization and the improved chaotic coupling mapping used in the joint encryption model are described in detail in Section 3.1 and Section 3.2 below.

### 3.1. Realization of AENN Randomization

There are two stages for the implementation of AENN random selection.

Stage 1: Construct the AENNs.

The designed AENN encoding network is shown in Figure 4. Two hidden layers are used to encode the 0-1 bit data into three floating-point numbers (float type) as output results. Encoding is carried out in bytes, meaning every eight bits of binary code are encoded into three floating-point numbers. Because the output floating-point is a 32-bit binary number with strong expressive force, and the input 8 bits are to the power of 2, it is lossless to represent the 8th power of 2 with the 32nd power of 2. According to the description in Section 2.2, eight bits of a byte belong to the input layer, and three floating-point numbers belong to the middle layer, also considered the output eigenvalues of the encoder. The three floating-point numbers output by the encoder cannot be directly transmitted as codes, and need to be quantized. The floating-point quantization process is shown in Algorithm 1. Its essence is to convert the normalized floating-point number into binary representation using the algorithm of multiplying decimals by 2 to obtain integers. The results of the operation retain the finite precision of 8 bits, and finally, the 8 bits are combined into a byte as the quantized output.
**Algorithm 1** Decimal to byte**Input:** A normalized floating-point number *X* and the max bit length L=8 (a byte) for conversion.**Output:** Byte *R*. 1: i←0 2: S← EMPTY 3: **while**
X≠0
&&
i<L
**do** 4:    v=int(X*2) //‘int’ means the function of integer conversion. 5:    X=X*2−v 6:    S=S+string(v) 7:    i=i+1 8: **end while** 9: *R* = byte(*S*) // Binary decimal parts converted into bytes10: **return** *R*


To correctly learn the input information, the hidden layer needs to have enough neurons, and gradually reduce the number of neurons in the encoding process to extract data features. In this paper, the number of nodes in hidden layer 1 adopts about 64 nodes, which is more reasonable than 8 input neurons. The number of nodes in hidden layer 2 is about 32 nodes. Such a network structure can both correctly express the input information and quickly reduce the size of neurons.

Another important component of the AENN coding network is the activation function. Choosing an appropriate activation function is very important for neural network training. We used the ReLU function as the activation function in hidden layer 1 and hidden layer 2. This is because the derivative of the ReLU function is a piecewise linear function, and it is not easy to cause the gradient to disappear when there is no saturation problem (the function slope is called function saturation when it is 0). The two hidden layers, 1 and 2, have more neurons, so using the ReLU function can avoid a lot of exponential operations and improve the calculation efficiency. As the middle layer of the encoder output, it must have strong expressive force to better express the characteristics of input data. Therefore, we do not use the ReLU function in the middle layer but use the sigmoid function, which is fully derivable as the activation function. Using this combined activation function scheme both increases the training speed and prevents the majority of neurons from dying (the output is 0).

The AENN reconstruction network is shown in Figure 5, which adopts a network structure that is symmetrical to the encoding network. The output characteristic value of the AENN encoder is used as the input of the decoder. In Algorithm 1, the eigenvalues (three floating-point numbers) of the encoder output have been quantized into three bytes, so the bytes need to be dequantized before being input into the decoder. The inverse quantization process is shown in Algorithm 2. First, the byte is converted into a bit sequence, which is used as the binary weight of the decimal part, and then it is converted into a floating-point number by the method of “weighted addition”. Then, taking the three floating-point numbers as the input, eight floating-point numbers as outputs are obtained. Then, the outputs are rounded and forcibly converted to bit data, i.e., eight bits. Hidden layer 3 and hidden layer 2 have the same number of nodes, and hidden layer 4 and hidden layer 1 have the same number of nodes. Similarly, in order to speed up the training and prevent the overfitting problem, this paper does not use the activation function in hidden layer 3, and the ReLU function is used as the activation function in hidden layer 4 and the output layer.

Stage 2: Random selection of AENN.

AENN has an auto-learning characteristic, i.e., the training process is the process of learning automatic encoding and decoding, and the weight parameters obtained can be regarded as the key of the encryption algorithm. For a single AENN coding network, there is an equivalent key. This is because each training of the neural network will almost obtain different weight parameters, i.e., there are many equivalent keys. Therefore, the security performance of a single AENN-encoding network is low. The premise of obtaining the equivalent keys is to obtain enough input and output coding pairs with strong correlation and to establish a similar network for training. The stronger the randomness of the autoencoder, the higher the security. To enhance the randomness of the encoder, multiple AENN networks can be used for selective coding. Following the simple and effective principle, this paper takes four AENN networks as examples for selective coding. The input and output coding pairs obtained by the diversified AENN structure have weak correlation, so the equivalent key cannot be obtained, and even they cannot be trained effectively. A variety of AENN structures greatly increase the key space and nonlinear complexity. To make the AENN have greater differentiation and reduce the correlation, the number of nodes is fine-tuned in the two hidden layers and four differentiated AENNs are obtained, as shown in Table 2.
**Algorithm 2** Byte to decimal**Input:** Byte *R***Output:** A floating-point number *X*.1: i←0, X←02: *S* = bin(*R*) //Byte is converted to binary3: **for**
i=0 until len(*S*) **do**4:    v=2(−i−1)*int(S[i])//‘int’ means the function of integer conversion.5:    X=X+v6: **end for**
7: **return** 
*X*


The random selection of the AENN structure depends on chaotic sequences. This is because the chaotic sequence has high randomness and unpredictability, and it is easy to realize the synchronization between the two sides of the communication. The AENN selection sequence is generated using ILM, and the initial value entered is marked as $X_{0}$.

The chaotic sequence obtained by ILM is a series of random numbers $\left\{x_{k}\right\}$ in the interval (0,1), with the form of {0.15537184388449532, 0.21636839614257042, 0.8841541779975159, 0.15110890584401204, 0.5299402990385715, ⋯}. To be used in encryption, we must quantize the chaotic sequence. The quantization method adopts the multi-level uniform quantization introduced in Section 2.1. If the *L* value of Equation (Equation 4) is taken as 2, then the real number sequence $\left\{x_{k}\right\}$ is converted to the integer sequence $\left\lfloor 2^{2}x_{k}\right\rfloor$. Because $x_{i}\in \left(0,1\right),i=0,1,2,\cdots$, the range of values for integer sequences is {0,1,2,3}. The discrete integer value will determine the selection of AENN.

The randomly selected AENN encoding and decoding processes are shown in Figure 6. As the number of encoders increases, the hybrid coding structure becomes more complex. Therefore, choosing an appropriate number of encoders not only prevents the encoding results from being easily learned, but also improves a reasonable operation speed. Here, four encoders are chosen to mix and encode in parallel because the security is enhanced without being too complicated. Among them, one byte is encoded and quantized into three bytes, and the encoding space of a single byte is $2^{24}$.

Here, the encoding process is the encryption process. Consequently, the neuron parameters in the encoder are all keys. Still, since the attacker can perform brute force cracking by enumerating the encoding table, the key size of the encoder can be accurately described only by using the encoding space. Moreover, the generation of the randomization selection sequence depends on the ILM. Therefore, $X_{0}$ is one of the encryption keys, which needs to be shared by both parties before using AENN coding and decoding.

### 3.2. A New Type of Coupled Chaos Mapping

In order to increase the complexity of chaos maps, different low-dimensional chaos can be used for coupling. The coupling mapping of ICM and ILM, introduced in Section 2.1, is used to improve the complexity.

Figure 7 shows the cross-mapping process of two types of chaos, comprising two branches. The cross-mapping process is as follows:In the left branch, set the control parameter $\mu_{1}$ and initial value $U_{1}$ of ICM, and obtain a real number $y_{1}$ after the ICM operation;In the left branch, set the control parameter a=1 of ILM, input the real number $y_{1}$ into ILM, and calculate to obtain a real number $y_{2}$ to complete the first cross-mapping of the left branch;If the number of cross-mappings n>1, the real number $y_{2}$ is input into the ICM of the left branch, and steps 1 and 2 are repeated until a real number $y_{2n}$ is output after *n* times of mapping and added to the real number sequence $f_{0}$;In the right branch, set the control parameter a=1 and the initial value $X_{1}$ of ILM, and obtain a real number $z_{1}$ after the ILM operation;In the right branch, set the control parameter $\mu_{2}$ of ICM, input the real number $z_{1}$ into the ICM, and calculate to obtain a real number $z_{2}$ to complete the first cross-mapping of the right branch;If the number of cross-mappings m>1, the real number $z_{2}$ is input into the ILM of the right branch, and steps 4 and 5 are repeated until a real number $z_{2m}$ is output after the mapping for *m* times and added to the real number sequence $f_{1}$;Every time the Y map and Z map output, the real numbers obtained are not only added to the sequences $f_{0}$ and $f_{1}$, but are also used as the initial values of the next iteration of their respective branches. The two branches iterate separately until real number sequences $f_{0}$ and $f_{1}$ of sufficient length are obtained;Set the L value in the quantization Equation (Equation 4), quantize $f_{0}$ and $f_{1}$ into two bit sequences, respectively, and perform the XOR operation to obtain the bit sequence $f_{2}$.

It should be noted that ILM is a full map at a=1, while ICM is a full map in the range of μ, so control parameter a of the ILM is set to 1, and both the control parameters $\mu_{1}$ and $\mu_{2}$ of ICM can be used as keys. In Figure 7, the left branch ICM and ILM form a new map, the Y map, and the right branch ILM and ICM form a new map, the Z map. By setting different cross-mapping times, *n* and *m* of the Y map and Z map to different values, respectively, and calculating the LE, as shown in Figure 8, it can be seen that the LE of the new map is greater than that of the single ILM and ICM when μ is equal (Figure 1d and Figure 2d), showing that cross-mapping improves chaos performance. When *n* and *m* are greater than or equal to 2, the LE of the new map tends to be stable, and is greater than the LE when *n* and *m* are equal to 1. Therefore, the number of cycles *n* and *m* of the new chaos is set to 2, which not only improves the chaos performance but also reduces the calculation amount.

The cross-mapping method effectively improves the chaos performance. We list the maximum LE, initial conditions, and range of parameters of the new chaos proposed in recent years, as shown in Table 3. It can be seen that the cross-mapping proposed in this paper has a very high LE, and through cross-mapping, it not only has multiple chaotic initial values, but also increases the number of chaotic control parameters.

Since the correlation analysis of sequences can clearly describe the linear relationship between two sequences, we use the Pearson correlation coefficient to calculate the similarity of sequences generated by chaotic cross-mapping at different times. The equation for calculating the Pearson correlation coefficient is [55]:(11)$$\rho_{X,Y}=\frac{COV(X,Y)}{\sigma_{X}\sigma_{Y}}=\frac{E(XY)-E(X)E(Y)}{\sigma_{X}\sigma_{Y}}$$

The value range of $\rho_{X,Y}$ is [−1,1]. When the value is 1, it means that there is a complete positive correlation between the two series; when the value is −1, it means that there is a complete negative correlation between the two series; when the value is 0, there is no linearity between the two series relation. To test the correlation of the sequences generated by chaotic cross-mapping, the Pearson correlation coefficient of the sequences generated by chaotic cross-mapping at different times is calculated and illustrated in Figure 9 and Figure 10. It can be seen from Figure 9 and Figure 10 that the autocorrelation coefficient and cross-correlation coefficient of the sequence are both close to 0 and evenly distributed, indicating that the sequence has no correlation.

## 4. Encryption and Decryption Algorithm of the Joint Encryption Model

### 4.1. Encryption Algorithm

The purpose of this system is to achieve efficient encrypted communication, and the plaintext messages we transmit are processed and analyzed in binary format. For each frame of data, we first encode it according to bytes to obtain the corresponding floating-point data, and then quantify the floating-point data to obtain the binary data. Finally, the chaotic map introduced in Section 2.1 is used to couple the data, as per the method described in Section 3.2, to generate a chaotic sequence, which is combined with the encoded binary data using “bitwise XOR”. The result of this diffusion is transmitted as ciphertext.

The encryption process is described as follows:According to the byte length of plaintext *P*, an equal-length chaotic sequence *X* ($X_{i}\in \left\{0,1,2,3\right\}$) is generated by ILM;According to the chaotic sequence *X* of step 1, the corresponding AENN (from 0 to 3) is selected to encode the plaintext bytes to obtain the floating-point sequence *F*;Quantize the floating-point number sequence *F* according to Algorithm 1, and convert the quantization result into a bit sequence *B*;The real number sequence is generated by the cross-mapping of ICM and ILM, and the binary chaotic sequence *H* with the same length as the bit sequence *B* is intercepted after quantization;Conduct the bitwise XOR operation between sequences *H* and *B* to obtain the ciphertext *C*.

The entire encryption process is shown in Figure 11.

### 4.2. Decryption Algorithm

The decryption process is the inverse process of the encryption, as far as symmetric encryption is concerned. The control parameters and initial conditions of the chaotic mapping of the decryption side should be consistent with that of the encryption side, and then the ciphertext should be processed according to the inverse process of the encryption algorithm. The steps in the decryption process are described below:Set the same parameters and initial values as encryption, generate a real number sequence by cross-mapping of ICM and ILM, and intercept a binary chaotic sequence *H* with the same length as ciphertext *C* after quantization;Ciphertext *C* and binary chaotic sequence *H* are operated by bit-wise XOR to obtain binary sequence *B*;The binary sequence *B* is converted into the byte sequence, and the inverse quantization operation is carried out to obtain the floating-point sequence $F^{\prime }$ (there is a slight difference with *F* in Section 4.1, which is caused by the accuracy of quantization and does not affect the decoding result.)The sequence *X* is generated by ILM, which is used to select AENN (from 0 to 3). In addition, the length of *X* is 1/3 of the floating-point sequence $F^{\prime }$.Use the selected AENN to reconstruct the corresponding floating-point numbers in $F^{\prime }$, so that the original byte is decrypted out. Thereby, the entire plaintext *P* could be obtained.

The entire decryption process is shown in Figure 12.

It should be noted that the above encryption and decryption processes are only described for one frame of data. In secret communication where plaintext is divided into different frames, the initial conditions $X_{0}$, $X_{1}$, and $U_{1}$ need to be changed, respectively, to resist common password attacks, which will be described in Section 5.

## 5. Safety Performance Analysis

A good encryption algorithm must have enough key space, high key sensitivity, strong pseudo-randomness, and can resist common attacks. In order to evaluate the security of the proposed joint encryption model, we will analyze the security of the joint encryption model in the key space, key sensitivity, resistance to selective plaintext attacks, and resistance to statistical attacks.

### 5.1. Key Space Analysis

Although the weight parameter of the AENN is equivalent to the key of the encryption algorithm, the neural network does not need to be simulated, while the coding space is used in the enumeration process. In this paper, when coding, the inputs of 8 bits are converted into three floating-point numbers, which are converted into three bytes for transmission, so the coding space of a single byte is $2^{24}$. Because there are 256 different bytes, the total coding space is $(2^{24})!$. Therefore, the AENN proposed in this paper has a huge key space.

For a chaotic sequence, both the initial value of the chaotic system and the control parameters of the chaotic system can be used as the keys. The ICM and the ILM are introduced in Section 2.1, and then we couple the improved maps to obtain a new chaos map with lower computational complexity and a sufficiently large key space.

Among them, the neural network selection uses the sequence generated by ILM, which has the initial value $X_{0}\in \left(0,1\right)$ and its variable step size is $10^{-16}$. Thus, we have $S_{X_{0}}=1\times 10^{16}$. ICM and ILM are used as sequence generators for coupled chaos mapping. For ICM, parameters $\mu_{1},\mu_{2}\in \left(0,10\right]$, and $\mu_{1},\mu_{2}$ have variable step sizes $10^{-16}$, and the initial values $U_{1}\in \left[0,1\right]$ and $U_{1}$ have variable step sizes $10^{-16}$. Thus, we have $S_{\mu_{1}\mu_{2}}=1\times 10^{34}$ and $S_{U_{1}}=1\times 10^{16}$. For ILM, its initial value $X_{1}\in \left(0,1\right)$ and $X_{1}$ have variable step sizes $10^{-16}$. So $S_{X_{1}}=1\times 10^{16}$ is obtained. As the generating factor of chaotic initial conditions, the communication times *N* are included in the enumerated range of initial values above. Thus, the key space generated by using all these chaos sequences is as follows: $S_{1}=S_{X_{0}}S_{\mu_{1}\mu_{2}}S_{U_{1}}S_{X_{1}}=1\times 10^{82}$.

The advantage of joint encryption is not only to increase the complexity of the algorithm, but also to expand the key space. And the total key space of the joint encryption model proposed by this paper is as follows: $S=S_{0}S_{1}>\left(2^{24}\right)!\times 10^{82}\gg 2^{100}$, which is enough to resist exhaustive attacks.

### 5.2. Key Sensitivity Analysis

Key sensitivity is an important factor in the evaluation of encryption algorithms, i.e., a small change in the key can cause sufficient changes in the plaintext. From a statistical point of view, the change rate must be close to 50%. The chaos map is highly sensitive to control parameters and initial conditions. The initial conditions of the chaotic map used in this paper are all double-precision floating-point numbers. For changes in the initial conditions less than $10^{-16}$, the chaotic map will produce the same result. Therefore, at the premise of keeping the other conditions and parameters unchanged, the initial condition of ILM is modified 39 times with each step size $10^{-16}$. And the bit-changing rate of the last nine times is calculated, as shown in Table 4.

As can be seen from Table 4, the bit change rate of the last nine encryptions is close to 50%. It shows that ILM satisfies key sensitivity when the initial condition change is greater than or equal to $10^{-16}$. The same tests are conducted on the initial conditions of several other chaotic maps, and the results are consistent, which indicates that the proposed algorithm has key sensitivity.

### 5.3. Analysis of Resisting Chosen Plaintext Attack

The chosen plaintext attack consists of the eavesdropper obtaining the authority of part of the encryption machine, which heavily threatens the security of the ciphertext. Different from known plaintext attacks, chosen plaintext attacks can help crack the plaintext of the key for encryption analysis. In the known cracking methods, it is difficult to crack the initial conditions of chaos and the failure rate is high. However, in the case of the chosen plaintext attack, as long as the equivalent key can be obtained, other encryption data can be cracked.

The steps for a generally chosen plaintext attack with equivalent keys are:Choose a regular text or image to encrypt with an encryption machine to obtain ciphertext *C*;Plaintext *M* is diffused by conducting the XOR operation with the chaotic sequence $K^{\prime }$, then $K^{\prime }$ is the equivalent key of the algorithm. The equivalent key $K^{\prime }$ can be obtained by XOR between the ciphertext *C* and plaintext *M*:
$$K^{\prime }=C⊕M$$According to the obtained equivalent key $K^{\prime }$, an equivalent key ${\hat{K}}^{\prime }$ of a different length can be intercepted according to the ciphertext of a different length, and other plaintext information $M^{\prime }$ can be obtained:
$$M^{\prime }=C⊕{\hat{K}}^{\prime }$$

Aimed at the above methods of the chosen plaintext attacks, the traditional ways are:Calculating the initial conditions of chaotic mapping based on hash algorithms and plaintexts, which can resist differential attacks [23,27];Generating the initial conditions of chaotic mapping based on true random number generators [25].

The above two methods can achieve approximately one-time pad encryption, but they are also very costly. Both communication parties need to use a secure channel to share keys in real time or perform very complex key management. The resource consumption of the secure channel and the complicated key management leads to the reduction of the efficiency of encryption communication. To improve the efficiency of encryption, two methods are designed in this paper to eliminate the equivalent key $K^{\prime }$:Simultaneous replacement of initial conditions of chaotic mapping. Chaotic mapping has the principle of unpredictability and determinism, i.e., the chaotic iteration result cannot be predicted if the chaotic initial conditions are not known, while chaotic synchronous iteration produces the same result if the chaotic initial conditions are known. Therefore, after sharing the initial conditions once for both communication parties, the chaotic synchronous iteration produces the same iterative result and the iterative result is used as the initial condition for the next communication, showing that different communication frames have different initial conditions;Using other synchronizable factors, such as the generation factors of the initial conditions. In continuous communication, using chaotic synchronous iterations to change the initial conditions is equivalent to using the same initial conditions. Therefore, adding other easily synchronizable factors, such as the number of communications and real-time time, can further increase encryption security. In this paper, the reciprocal of the number of communications is used to make subtle changes to the initial conditions, thus eliminating the equivalent key.

The proposed method only requires the communication parties to share the initial conditions once at the beginning of the communication. It can continuously encrypt different data frames without sharing the key in real time while the communication parties perform error-free encryption transmission. This method, combined with the sensitivity of chaotic mapping to initial conditions, approximately achieves the effect of one-time pad encryption and has the ability to resist the chosen plaintext attacks.

### 5.4. Analysis of Resisting Statistical Attack

Using statistical analysis tools to perform statistical analysis on ciphertext is one of the most commonly used methods in ciphertext-only attacks. Therefore, the randomness test of the ciphertext can verify the ability of the encryption algorithm to resist statistical analysis. NIST SP800-22 [56] is the NIST statistical test suite provided by the National Institute of Standards and Technology of America, which contains 15 of the most commonly used statistical test methods. Each statistical method can obtain different test-result credibility, according to different *p*-values. The default *p*-value of the test suite is 0.01, which means that the test result has a 99% confidence level.

We use the cross-mapping in Section 3.2 to generate real number sequences, set the parameter L=8 in Equation (Equation 4), quantize each real number into eight bits, splice them, intercept the bit sequence with the size of 1 Gb, and apply NIST test to it. The *p*-values for the 15 tests are shown in Table 5. As can be seen from Table 5, all p values of the ciphertext are greater than 0.01, passing all the NIST test items successfully.

## 6. Experimental Results and Analysis

By using the joint encryption model in passive optical network communication, the encrypted communication experiment will be simulated in this section. The system diagram of the encrypted communication experiment can be seen in Figure 13.

### 6.1. Text Encryption Analysis

The proposed algorithm was implemented in Python, the software platform was PyCharm 2021 (Community Edition), the operating environment was Windows 10 Home Edition, the CPU specification was Intel(R) Core(TM) i5-10300H @ 2.50GHz, and the RAM size was 16GB.

Encrypting the same plaintext can verify the sensitivity of the algorithm. Here, we encrypt the same string “KKKKKKKK” as an example, and the result is shown in Table 6.

As shown by the experimental results in Table 6, the high sensitivity of the algorithm is demonstrated by the following two points:The same character encryption obtains a completely different ciphertext sequence;The same plaintext information obtains completely different results in each encryption.

### 6.2. Image Encryption Analysis

To further illustrate the security of the encryption algorithm for other types of data, the image is selected as the object of encryption, and we compare the evaluation data of the image encryption with other chaotic image encryption algorithms. Based on the purpose of the equivalence comparison, this paper only chooses the coupled chaos mapping to encrypt the image, in order to prove the security performance of the proposed coupled chaos mapping in image encryption.

“Lena” is often used as the object of image encryption. In this paper, the difference between the two initial values of the ILM in the coupled chaos mapping is set to $10^{-16}$, and other chaotic parameters and initial values remain unchanged. We use these two sets of parameters to encrypt Lena to obtain two encrypted images, as shown in Figure 14b,c.

The histogram reflects the statistical properties of the image pixels. Figure 15a–c represent the histograms of the original image and the two encrypted images, respectively. It can be seen from the histogram that the original image has certain pixel distribution characteristics, while the encrypted image presents a uniform distribution, and there are no similar characteristics between the histograms of the two encrypted images. This shows that the attacker cannot use the statistical characteristics of the image to analyze the encrypted image. Therefore, the algorithm proposed in this paper has the ability to resist statistical analysis in image encryption.

In addition to the histogram, many image encryption indicators can evaluate the image encryption security. In this paper, several commonly used evaluation criteria, such as the pixel number change rate (NPCR), unified average change intensity (UACI), information entropy, and encrypted image pixel correlation, are calculated, and different images are encrypted; the results are shown in Table 7.

NPCR and UACI are metrics used to evaluate the sensitivity of image encryption, and their ideal values are 99.6094% and 33.4635%, respectively [57]. Table 7 shows that the NPCR and UACI of the coupled chaos mapping proposed in this paper are very close to the ideal values, i.e., the sensitivity of the security key is strong enough. Information entropy is an index used to evaluate the randomness of an image. The larger the entropy value, the better the randomness. The ideal entropy value is eight. Table 7 shows that the entropy value of the coupled chaos mapping encrypted image proposed in this paper is 7.9993, which is very close to the ideal value. It means that the encrypted image pixel value is evenly distributed. The adjacent pixel correlation analysis is an index used to evaluate the similarity strength between adjacent pixels in an image. The smaller the correlation parameter, the weaker the correlation of pixels [58], and the encryption algorithm with high security has to make the correlation between pixels close to 0. Table 7 shows that the correlation parameter of the algorithm proposed in this paper is close to 0, i.e., the encryption algorithm effectively eliminates the correlation between pixels. Therefore, the encryption algorithm in this paper has high security.

In addition, we compare the encryption results of the “Lena” graph with those of other image encryption algorithms published in recent years, see Table 8. The results in Table 8 show that our proposed encryption algorithm can obtain similar or lower correlation results with other encryption algorithms only in the chaotic part, which can meet the requirements of image encryption for image correlation.

## 7. Conclusions

This paper addresses the short-period phenomenon of one-dimensional chaos, the shortcomings of low complexity, and the traditional one-time pad encryption that requires real-time key sharing to occupy a large amount of channel resources. This paper proposes a randomly selected AENN to improve the non-linear complexity and key space of the encryption algorithm. Its byte encryption is controlled by a chaotic sequence. Secondly, two low-dimensional ICM and ILM are used to couple into a new chaotic map, which can yield excellent chaotic dynamics performance, eliminate the short-period phenomenon, and increase the size of the key space. The iterative data of the chaotic map are used for another iteration, combined with the communication number to calculate a new initial condition of the chaotic map, so that the system can continuously change the key after sharing the key once. Finally, we propose a joint encryption model based on randomized AENN and a new coupled chaos mapping. Our encryption experiments have verified that this joint encryption model can save secure channel resources, resist common attacks, and thus, offer high security.

In future work, we will combine the advantages of different encryption algorithms to design a more reliable encryption scheme to improve data transmission security and realize the efficient and secure transmission of information. For example, chaotic encryption can be combined with the dynamic S-box, demonstrating good nonlinearity and differential uniformity, or a neural network with a multi-layer neuron structure, complex nonlinear relationships, etc. Thus, this addresses the issue where a single encryption method cannot provide comprehensive security assurance due to its own shortcomings, making data transmission more secure and efficient.

## Figures and Tables

**Figure 1 entropy-25-01153-f001:**
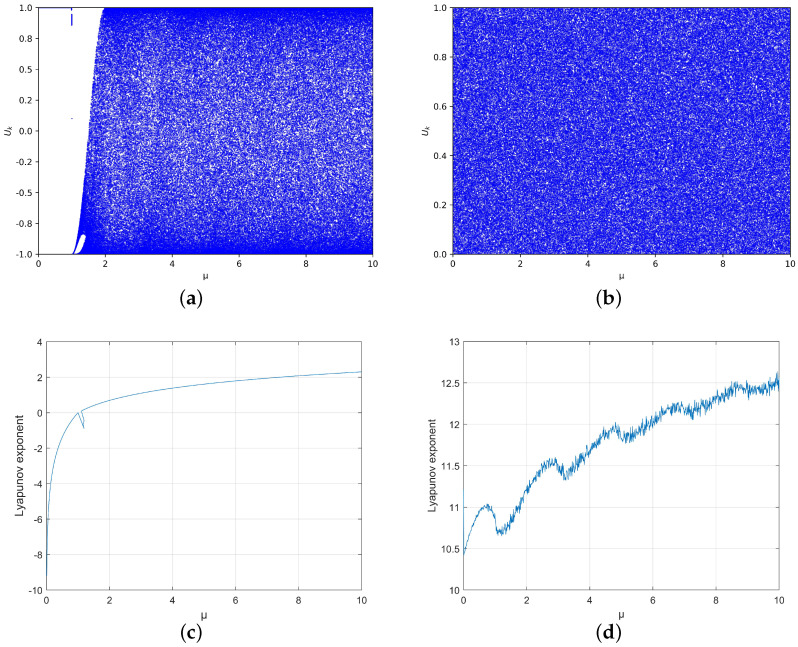
Bifurcation for (**a**) the Chebyshev map and (**b**) ICM and LE for (**c**) the Chebyshev map (**d**) and ICM.

**Figure 2 entropy-25-01153-f002:**
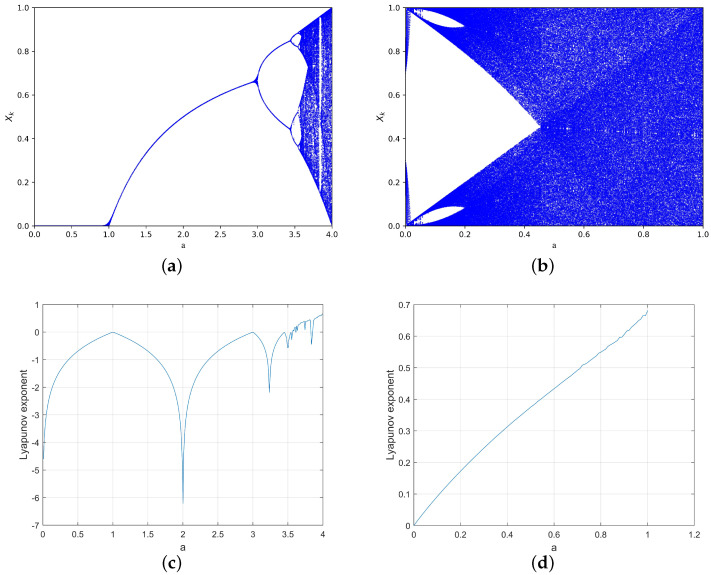
Bifurcation for (**a**) the logistic map and (**b**) ILM and LE for (**c**) the logistic map (**d**) and ILM.

**Figure 3 entropy-25-01153-f003:**
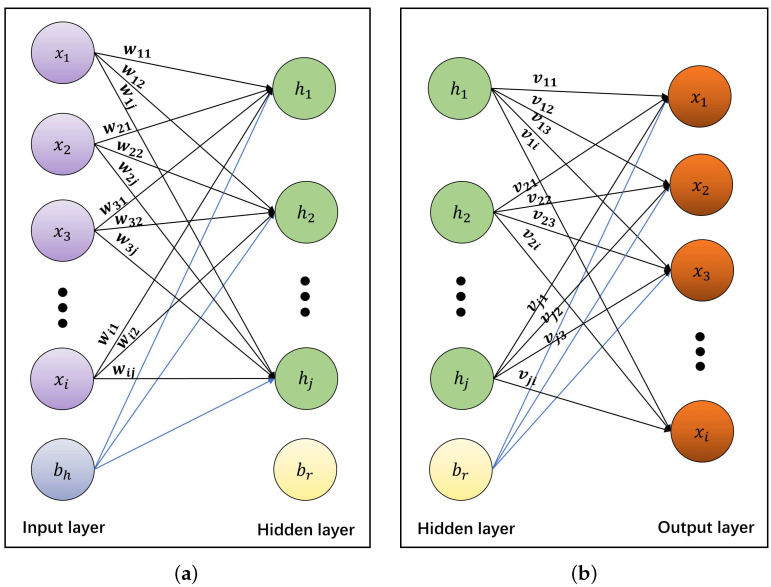
The AENN architecture. (**a**) Encoder, (**b**) decoder.

**Figure 4 entropy-25-01153-f004:**
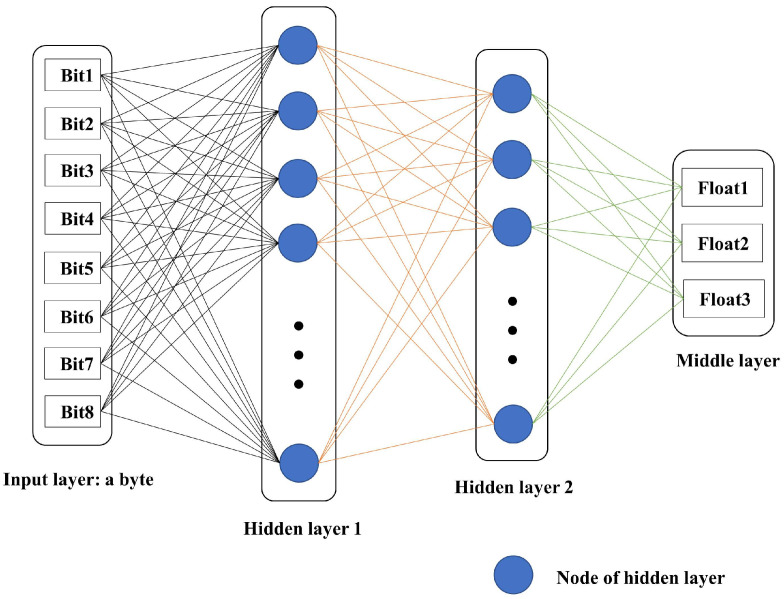
AENN encoding network with two hidden layers.

**Figure 5 entropy-25-01153-f005:**
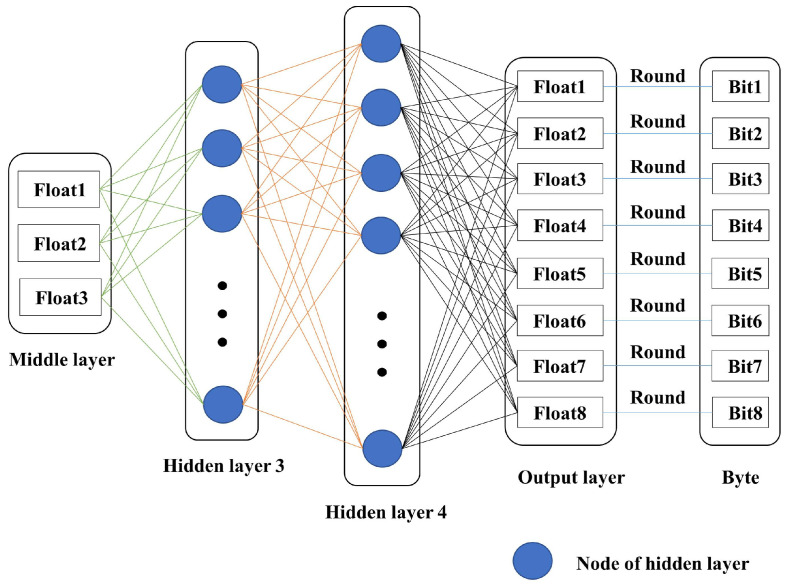
AENN reconstruction network with 2 hidden layers.

**Figure 6 entropy-25-01153-f006:**
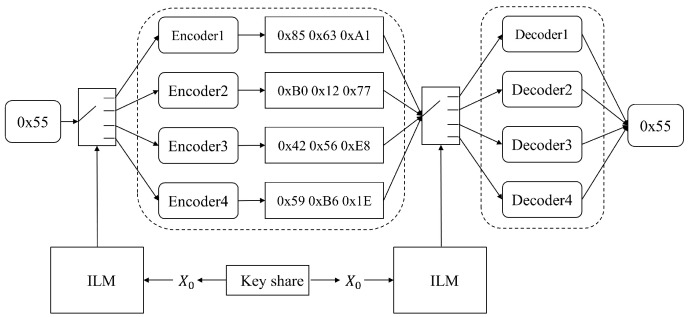
Four randomly selected AENN codecs.

**Figure 7 entropy-25-01153-f007:**
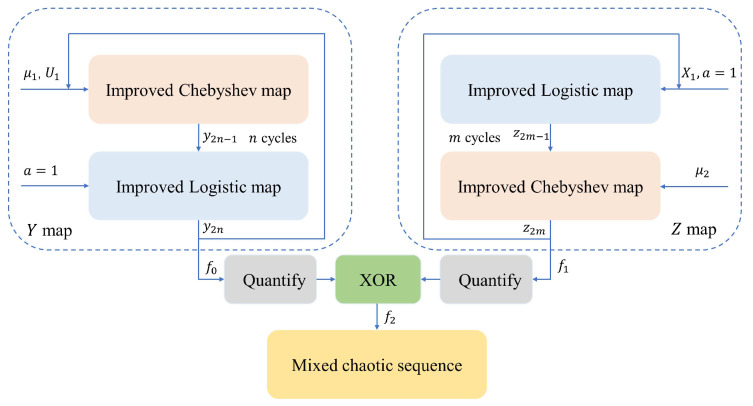
The schematic diagram of coupled chaos mapping.

**Figure 8 entropy-25-01153-f008:**
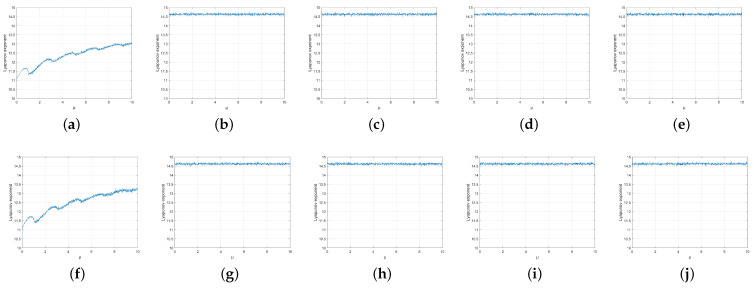
LE for (**a**) Y map (n=1) (**b**) Y map (n=2) (**c**) Y map (n=3) (**d**) Y map (n=4) (**e**) Y map (n=5) (**f**) Z map (n=1) (**g**) Z map (n=2) (**h**) Z map (n=3) (**i**) Z map (n=4) (**j**) Z map (n=5).

**Figure 9 entropy-25-01153-f009:**
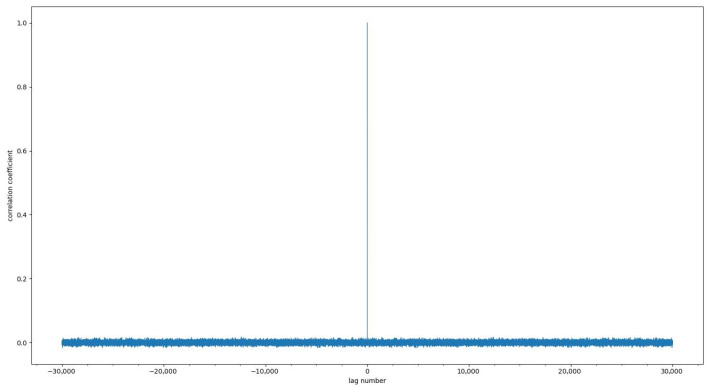
Autocorrelation of sequences.

**Figure 10 entropy-25-01153-f010:**
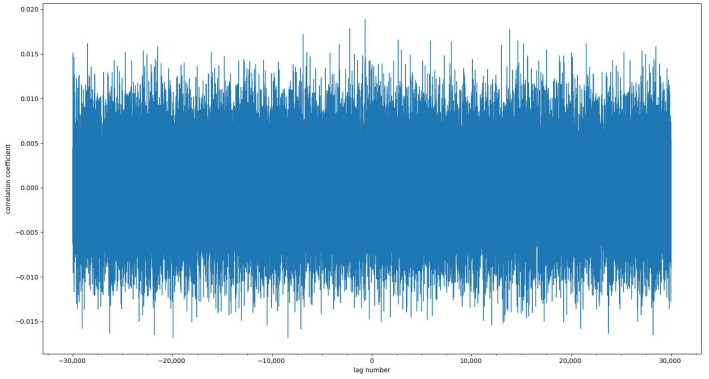
Cross-correlation of sequences.

**Figure 11 entropy-25-01153-f011:**
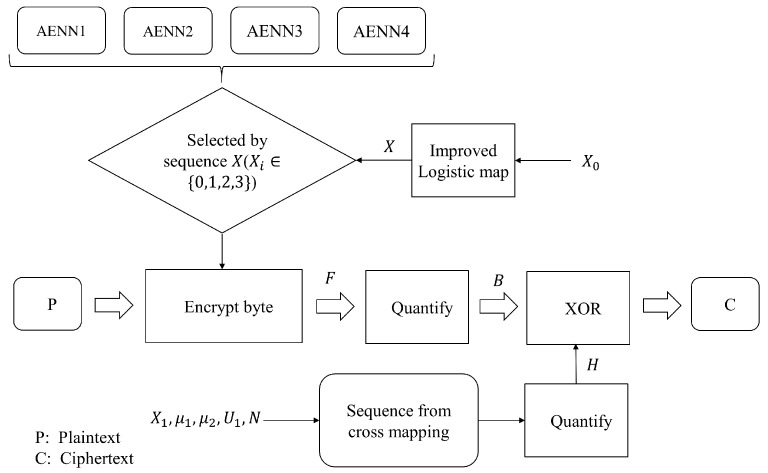
Schematic diagram of the encryption process.

**Figure 12 entropy-25-01153-f012:**
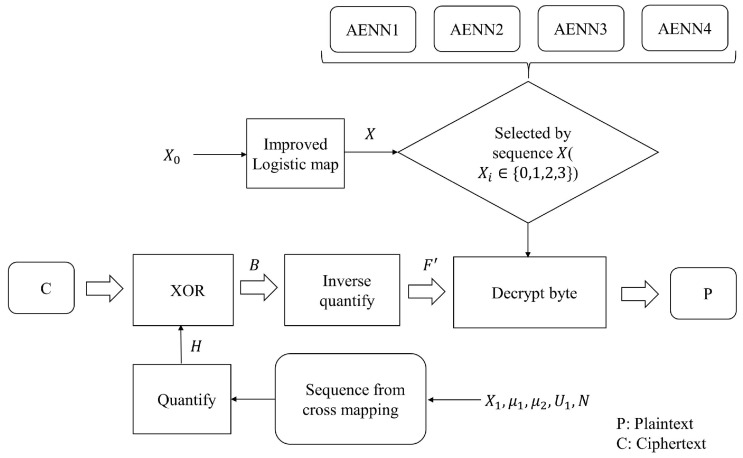
Schematic diagram of the decryption process.

**Figure 13 entropy-25-01153-f013:**
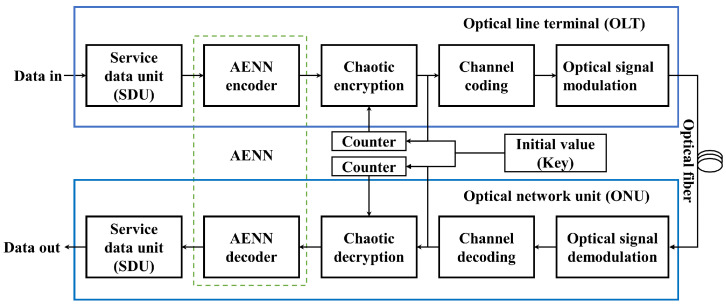
Experiment of the encrypted communication in passive optical networks.

**Figure 14 entropy-25-01153-f014:**
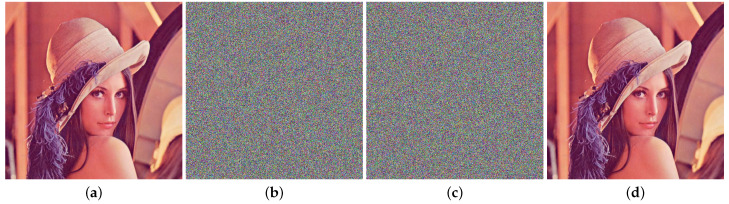
Encryption result (**a**) Original image of “Lena”, (**b**) encrypted image 1 of “Lena” (**c**), encrypted image 2 of “Lena”, (**d**) decrypted image of “Lena”.

**Figure 15 entropy-25-01153-f015:**
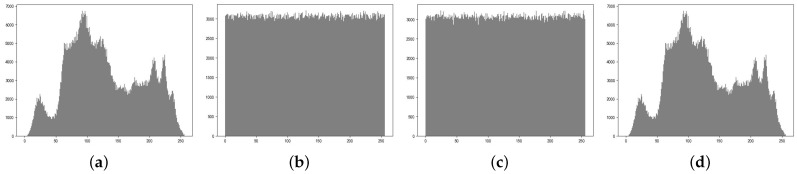
Histogram analysis, (**a**) histogram of plaintext “Lena”, (**b**) histogram of “Lena” encrypted image 1, (**c**) histogram of “Lena” encrypted image 2, (**d**) histogram of “Lena” decrypted image.

**Table 1 entropy-25-01153-t001:** Advantages and disadvantages of different encryption algorithms.

Encryption Algorithm	Advantages	Disadvantages
RSA	Good compatibility, high security	Slow encryption speed
AES	Multi-mode, efficient, large key space	Cannot hide the plaintext mode (e.g., image encryption contour is still in place)
S-box	Nonlinear transformation	Simple replacement
Chaotic encryption	Low complexity, Good diffusivity, large key space	Short period phenomenon, finite precision effect
Compression–encryption	Reduce data redundancy	Slow encryption speed
Encoding–encryption	Low complexity	Vulnerable to statistical attacks
Neural network encryption	Large key space, the trained encryption model is unique	Slow encryption speed

**Table 2 entropy-25-01153-t002:** The number of nodes in the hidden layer of the four AENN networks.

Neural Network	Nodes of Hidden Layer 1	Nodes of Hidden Layer 2
AENN1	63	34
AENN2	63	32
AENN3	64	32
AENN4	63	33

**Table 3 entropy-25-01153-t003:** Comparison with recent chaotic maps in encryption and cryptographic technology.

Chaotic Maps	Maximum Lyapunov Exponent	Initial Condition	Range of Parameters
Proposed	14.7612	$X_{0}\in \left(0,1\right)$, $U_{0}\in \left(0,1\right)$	$\mu_{1}\in \left(0,10\right]$, $\mu_{2}\in \left(0,10\right]$
ICM (Ref. [32])	12.6342	$X_{0}\in \left[0,1\right]$	μ∈(0,10]
ILM (Ref. [33])	0.6812	$X_{0}\in \left(0,1\right)$	a∈(0,1)
Ref. [50]	0.6453	$\left(X_{0},Y_{0}\right)=\left(1,-2\right)$	a∈(1,4.2]
Ref. [51]	2.5876	$X_{0}\in \left(0,1\right)$	μ∈[0.25,1]
Ref. [52]	2.0194	$X_{0}\in \left[0,1\right]$	μ∈(0,4]
Ref. [53]	6.7502	$X_{0}\in \left[0,1\right]$	a∈[5,+∞],λ∈[0,+∞]
Ref. [54]	1.3099	$X_{0}\in \left[0,1\right]$	p∈[0.25,0.5]

**Table 4 entropy-25-01153-t004:** The bit changing rate of the encrypted ciphertext for the last 9 times.

No.	1	2	3	4
Different rate	0.50213119	0.50123788	0.50019142	0.50161222
No.	5	6	7	8
Different rate	0.49940020	0.50231836	0.49907266	0.50212268
No.	9	max	min	mean
Different rate	0.50263740	0.50263740	0.49907266	0.50119156

**Table 5 entropy-25-01153-t005:** NIST test results.

Statistical Test	*p*-Value
Monobit	0.791932
Frequency within block	0.346868
Runs	0.612983
Longest run ones in a block	0.020584
Binary matrix rank	0.165968
Dft	0.544532
Non-overlapping template matching	0.999999
Overlapping template matching	0.853520
Maurer’s universal	0.209331
Linear complexity	0.041116
Serial	0.702299
Approximate entropy	0.702912
Cumulative sums	0.511410
Random excursion	0.108644
Random excursion variant	0.294263
Pass rate	15/15

**Table 6 entropy-25-01153-t006:** Encryption results of the same plaintext at different times.

Plaintext: “KKKKKKKK”	
**No.**	**Ciphertext**
1	∖xaf[J∖x0f&∖xbc∖xa8∖x12Mv∖xfd(∖∖∖t6∖x92q∖xb6∖x8b|∖xc2∖xb2∖∖∖x12
2	SSB∖xb3∖x92u∖xb1q∖xa5∖xbdo ∖xb6∖xc6∖xf9∖xad∖xd8∖x14%∖x96∖xf8]∖xf3
3	∖x01Ax∖xc6∖xc6∖xbc∖t∖xdc∖xef∖xecy∖xd3∖x15∖x9b∖x99∖xb0∖xf8∖x18∖xbe∖xeer
4	∖xee∖x9cI∖xc7∖x93∖xf3∖xc9d=∖x99∖xca.U∖x91QoP∖x8b∖x89∖x01∖xae∖xfe”x
5	∖xd0∖xc6∖xd8∖rRU∖xbe∖x16∖xbb∖x85vcEa∖x06∖x07$A ∖xb8∖xe0∖xdae]
6	I∖x15r∖x93∖x8a∖x06∖xf7X∖x93∖x04r∖xb7∖x8ct∖xa1∖xb4T∖xd5∖xf7f∖xfd∖xdaa∖xe1
7	∖xf9∖’X∖xce+∖x83Or∖x87∖xbc∖xa5∖xce∖xd0∖xc1f∖r<∖xc4∖x1e∖x94∖xe7∖xab∖xb2∖xa
8	∖xfa∖xcb∖xa5∖xe1∖xe8/QWv∖xcd,∖xea∖x8e∖xd1∖x1e∖xe2∖x06∖xd2∖xab∖xf1∖x7fXT
9	∖x81∖xd9∖xf3∖xd8∖xcaD∖x95h∖x86∖x17∖xe2∖x9e∖xd2M∖x84∖xd2∖xa0∖x05∖xd9-
10	∖x08J∖x86∖ri∖x94∖xd9;∖xc3∖x13P∖xc2∖xfeJ∖x05*∖x13∖x8b∖xeb∖x11∖x04∖x1f∖xd7
11	{∖x18∖xf0∖x89∖xe4∖xa8@∖x15∖x07lvk∖xbd∖x8c∖xd3∖x18∖x8c∖xdfNS∖xa0v=E
12	∖x02∖x00∖xae∖x02ITW∖x9a∖xfd∖xb5p∖xa6u∖x9b∖xab∖xd5∖xad#∖xe2w(∖xdd∖x93

**Table 7 entropy-25-01153-t007:** Entropy, correlation, NPCR, and UACI results of colored cipher images in the USC-SIPI database.

Image	Size	Average NPCR (%)	Average UACI (%)	Average Entropy	Average Correlation Analysis		
					Horizontal	Vertical	Diagonal
4.1.01	256×256	99.6058	33.3768	7.9973	0.0015	−0.0055	−0.0014
4.1.02	256×256	99.5972	33.4385	7.9971	−0.0064	0.0054	−0.0112
4.1.03	256×256	99.6072	33.4337	7.9971	0.0034	−0.0180	−0.0124
4.1.04	256×256	99.6063	33.5804	7.9973	−0.0059	−0.0055	0.0058
4.1.05	256×256	99.6043	33.4171	7.9972	−0.0158	−0.0010	−0.0015
4.1.06	256×256	99.6063	33.3703	7.9972	−0.0168	0.0168	0.0104
4.1.07	256×256	99.6150	33.5093	7.9970	−0.0059	−0.0054	−0.0230
4.1.08	256×256	99.6094	33.4748	7.9971	−0.0040	0.0161	−0.0024
4.2.01	512×512	99.6105	33.4755	7.9993	−0.0085	0.0235	0.0090
4.2.03	512×512	99.6174	33.4595	7.9992	−0.0064	0.0000	0.0231
4.2.05	512×512	99.5978	33.4694	7.9993	0.0049	−0.0136	−0.0073
4.2.06	512×512	99.6104	33.4671	7.9992	0.0030	−0.0072	−0.0136
4.2.07	512×512	99.6091	33.4615	7.9992	−0.0065	−0.0291	0.0068
House	512×512	99.6053	33.5365	7.9993	0.0127	0.0017	0.0019
Lena	512×512	99.6091	33.4623	7.9993	−0.0053	−0.0036	0.0135

**Table 8 entropy-25-01153-t008:** “Lena” graph encryption analysis of different chaotic image encryption algorithms.

Reference	Year	Key Space	Average NPCR (%)	Average UACI (%)	Average Entropy	Average Correlation Analysis		
						Horizontal	Vertical	Diagonal
Proposed		$10^{82}$	99.6091	33.4623	7.9993	−0.0053	−0.0036	0.0135
Ref. [59]	2023	>2${}^{100}$	99.61	33.49	7.9975	0.0007	0.0022	0.0085
Ref. [60]	2023	>$2^{100}$	99.6239	33.5290	7.9993	−0.0011	−0.0017	−0.0001
Ref. [61]	2022	>$2^{100}$	99.62	33.49	7.9993	−0.0018	−0.0039	−0.0001
Ref. [27]	2021	$10^{112}\times 4^{65536}$	99.60	33.38	7.9974	0.0105	−0.0025	0.0003
Ref. [24]	2021	>$2^{100}$	99.6125	50.0256	7.9992	0.0060	−0.0209	0.0055
Ref. [23]	2020	$2.9645\times 10^{149}$	99.5956	33.4588	7.9972	−0.0021	0.0009	0.0003
Ref. [62]	2019	$1.1\times 2^{377}$	99.6098	33.4707	7.9993	0.0125	−0.0174	−0.0065
Ref. [63]	2018	$10^{94}$	99.5999	33.3848	7.6635	−0.0041	0.0016	0.0021
Ref. [64]	2018	$1.9\times 2^{426}$	99.6128	33.4621	7.9998	0.0002	0.0004	0.0002

## Data Availability

The Mathematica notebooks and datasets used and/or analyzed during the current study are available from the corresponding author upon reasonable request.

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
