# Peer review of "Joint Encryption Model Based on a Randomized Autoencoder Neural Network and Coupled Chaos Mapping"

_entropy, 2023, doi:10.3390/e25081153_

Round 1
Reviewer 1 Report
Dear authors and editors
This work considers the problem of encryption through the combination of a neural network and a chaotic mapping.
I found the work very interesting and enjoyable to read, and I believe it holds merit to the scientific community.
Upon reading it though, I found several parts that required clarification, as they were hard to understood. To me, it is highly important to consider the reproducability of a work, since this is the only way other readers can replicate and build upon the results. Thus, I would advise the authors to keep this in mind when preparing their revisions.
Overall, I would advise for major revisions for this work. My comments can be found below.
--In Fig. 1 there is no reason for the bifurcation diagram of the original chebysev map to stop at 4. You can expand it up to the parameter value 10.
--For both the maps that you propose, it is essential that a Lyapunov exponent diagram be included.When analyzing chaotic maps, the bifurcation diagram is not enough to claim chaotic behavior. There must be an LE diagram as well to verify this.
--Moreover, claiming that one map is better than the other, is not straightforward. There are many aspects to which we can consider one map 'better'. For example, one such criterion can be a wider range of chaotic behavior. Another one can be the increase in the max LE value achieved. So I would suggest expanding your discussion here about the two maps.
--Building upon this comment, i believe the second map you propose, the ILM, is really not that better than the logistic map. I mean, the range only goes up to 1 for the parameter, and the chaotic region is just like in the chaotic map, very small. So i would say there really is not much improvement here.
--For figure 2, i would suggest reduce the stepsize for the ILM map, to obtain higher detail.
--line 126, you use z_j, but the figure 3 has h_j as a notation.
--lines 223-226, so what activation do you use, relu or sigmoid , i got confused here.
--for the floating point transformation at the output layer, how is the decimal part decided. Here again, please think of reproducibility, write this section so that any reader will be able to replicate this, include all the required details.
--line 263 'got'?
--line 261, again not clear how the transformation to integers is performed.
--figure 6, again the use of x0 in different parts requires clarification.
--section 3.2 is very unclear, the algorithm part i mean. It must be rewritten. The mention of f0 and f1 is very unclear and confusing. Please clarify this and make absolutely clear what is the process, and how each map is used.
Figure 10, with the corresponding analysis in the text is very confusing. How exactly are the two sequences related? If they are just 2 random sequences obtained from the maps, it is no surprise that the correlations are zero. WHat would be more itneresting, would be to see 2 sequences obtained from almost identical initial conditions, what is their correlation?
--line 324 and on, how exactly do you obtain the chaotic map in binary format? what is the Nist tests later tested for? what sequence? This is unclear.
--ALgorithm 1, i got very confused about the process. PLease describe it better.
--the correlation comparison is unfair. In most works, the authors choose just some random pixels, so all the correlation results are different each time. This must be mentioned in the image
Overall, i believe the way the nn and the chaotic map are combined required clarification.
The english is fine, only some proofread is required
Reviewer 2 Report
A joint encryption model based on randomized AENN and improved chaotic coupling map is proposed. the author's exploration of applications in imaging encryption contains some practical meanings. Results are correct and interesting. However, to improve the quality and presentation of the paper, the authors are suggested to address the following comments
1.The joint encryption model based on randomized AENN and improved chaotic coupling map is studied. I think some references on chaotic systems should be considered in the introduction, such as, IEEE Transactions on Circuits and Systems I: Regular Papers, 2023, DOI: 10.1109/TCSI.2023.3276983, Applied Mathematical Modelling, 2023, 121: 463-483, IEEE Transactions on Cybernetics, 2023, 10.1109/TCYB.2023.3267785.
2.The AENN proposed by the authors is very interesting. Specifically, how does AENN perform the encoding operation?
3.How are the initial values and parameters needed for the chaotic coupling map determined?
4.To verify the security of the proposed algorithm, the authors should also introduce more testing analysis, such as: correlation analysis, robustness analysis.
Please carefully check and modify whether the labels in the figure are written in a standardized manner, as well as the sorting and citation of references.
Reviewer 3 Report
The following are my major comments:
1. The authors claimed that they addressed the short period and low complexity of one-dimensional chaos. However, I could not find any comparative analysis of the proposed scheme with the existing schemes in this regard.
2. In general, I strongly suggest increasing the scope of comparison and do compare key spacing and other security factors with the existing schemes.
3. Similarly, please include the results of more images in the comparison table.
4. There are several other mathematical structures such as Elliptic curves that have longer periods and key spacing. I suggest also citing such schemes and may consider them for comparison. The following papers are for reference and are not necessary to cite them:
A Novel Image Encryption Scheme Based on Elliptic Curves and Coupled Map Lattices
Towards Provably Secure Asymmetric Image Encryption Schemes
A Novel Image Encryption Scheme Based on ABC Algorithm and Elliptic Curves
A new RGB image encryption algorithm based on DNA encoding and elliptic curve Diffie–Hellman cryptography
5. The structure of the sentences must be improved.
Round 2
Reviewer 1 Report
I believe all the questions were properly addressed.
Reviewer 2 Report
It is OK now after the last revision.
OK
Reviewer 3 Report
Thank you for revising the manuscript by following my comments.